# The LAMMER Kinase MoKns1 Regulates Growth, Conidiation and Pathogenicity in *Magnaporthe oryzae*

**DOI:** 10.3390/ijms23158104

**Published:** 2022-07-22

**Authors:** Lin Li, Xue-Ming Zhu, Jia-Qi Wu, Na Cao, Jian-Dong Bao, Xiao-Hong Liu, Fu-Cheng Lin

**Affiliations:** 1State Key Laboratory for Managing Biotic and Chemical Treats to the Quality and Safety of Agro-Products, Institute of Plant Protection and Microbiology, Zhejiang Academy of Agricultural Sciences, Hangzhou 310021, China; 21616143@zju.edu.cn (L.L.); 11816090@zju.edu.cn (X.-M.Z.); baojiandong@gmail.com (J.-D.B.); 2Zhejiang Provincial Key Laboratory of Biometrology and Inspection & Quarantine, College of Life Sciences, Jiliang University, Hangzhou 310018, China; 15605398347@163.com; 3College of Biotechnology, Tianjin University of Science & Technology, Tianjin 300457, China; cn19970819hao@163.com; 4State Key Laboratory for Managing Biotic and Chemical Treats to the Quality and Safety of Agro-Products, Institute of Biotechnology, Zhejiang University, Hangzhou 310058, China; xhliu@zju.edu.cn

**Keywords:** *Magnaporthe oryzae*, LAMMER kinase, pathogenicity, conidiation

## Abstract

*Magnaporthe oryzae* is an important pathogen that causes a devastating disease in rice. It has been reported that the dual-specificity LAMMER kinase is conserved from yeast to animal species and has a variety of functions. However, the functions of the LAMMER kinase have not been reported in *M. oryzae*. In this study, we identified the unique LAMMER kinase MoKns1 and analyzed its function in *M. oryzae*. We found that in a *MoKNS1* deletion mutant, growth and conidiation were primarily decreased, and pathogenicity was almost completely lost. Furthermore, our results found that MoKns1 is involved in autophagy. The Δ*Mokns1* mutant was sensitive to rapamycin, and MoKns1 interacted with the autophagy-related protein MoAtg18. Compared with the wild-type strain 70−15, autophagy was significantly enhanced in the Δ*Mokns1* mutant. In addition, we also found that MoKns1 regulated DNA damage stress pathways, and the Δ*Mokns1* mutant was more sensitive to hydroxyurea (HU) and methyl methanesulfonate (MMS) compared to the wild-type strain 70−15. The expression of genes related to DNA damage stress pathways in the Δ*Mokns1* mutant was significantly different from that in the wild-type strain. Our results demonstrate that MoKns1 is an important pathogenic factor in *M. oryzae* involved in regulating autophagy and DNA damage response pathways, thus affecting virulence. This research on *M. oryzae* pathogenesis lays a foundation for the prevention and control of *M. oryzae*.

## 1. Introduction

*Magnaporthe oryzae* is the causal agent of rice blast, a devastating disease affecting rice plants worldwide [1]. The rice blast fungus *M. oryzae* has become an important model fungus for studying the development of pathogenicity and microbe−plant interactions of pathomycete. The appressorium of *M. oryzae* is a special structure produced by conidia germ tubes or hyphal tips. The formation and development of the appressorium is crucial for plant surface penetration [2]. Many signaling or protein transport pathways, including protein kinases, DNA damage response and autophagy, are important for conidia germination, appressorium penetration and the extension of invasive hyphae of *M. oryzae* during the infection process [3,4,5].

Kinases capable of phosphorylating all three amino acid residues (serine/threonine/tyrosine) are called as dual-specificity kinases [6]. LAMMER kinases contain a conserved motif, ‘EHLAMMERILG’, in the subdomain X of the kinase catalytic domain [7]. LAMMER kinases are dual-specificity kinases. *KNS1* exhibits 36% identity to the *CDC2* gene product of *Schizosaccharomyces pombe* and 34% identity to the *CDC28* gene product of *Saccharomyces cerevisiae*. In yeast, it was initially thought that the *KNS1* gene is not essential for cell growth and a variety of other cellular processes [8]. Additional functions of LAMMER proteins have been discovered. In *S. cerevisiae*, the LAMMER kinase Kns1 is involved in filamentous growth and in the TOR signaling pathway [9]. In *Schizosaccharomyces pombe*, the LAMMER kinase Lkh1 is involved in filamentous growth, asexual flocculation, the oxidative stress response by regulating the expression of antioxidant genes and the regulation of Tup transcriptional repressors [10,11,12,13]. In *Drosophila*, the LAMMER kinase DOA plays a role in cell structure maintenance, differentiation and sex determination [14]. In *Aspergillus nidulans*, the LAMMER kinase Lkh1A affects vegetative growth, asexual and sexual development, and cell wall biogenesis [15,16]. In *Candida albicans*, the LAMMER kinase Kns1 is involved in dimorphic transition, cell wall integrity, response to DNA replicative stress and adherence to the host cell surface [17].

Autophagy is an evolutionarily conserved intracellular waste treatment and recycling process. In filamentous fungi, autophagy plays an extremely important function. In our previous research, we obtained the Δ*Moatg1–10*, Δ*Moatg12*, Δ*Moatg14–*Δ*Moatg16* and Δ*Moatg18* mutants, which completely lack pathogenicity [18]. The TOR kinase complex is an evolutionarily conserved protein kinase coordinating nutrients and energy in the regulation of cell proliferation, cell cycle, cell wall integrity and cell metabolic activities by interacting with different subunits [19]. In *M. oryzae*, it has been confirmed that TOR signaling functions in controlling the appressorium formation. TOR pathways are involved in modulating autophagy in coordination with the AMP-activated protein kinase ABL1 [5]. The vacuolar protein Imp1 maintains the integrity of the biotrophic interface membrane and promotes invasive hyphal growth by TOR [20]. The VASt domain-containing protein MoVast1 acts as a novel autophagy inhibition factor that modulates the activity of MoTOR [21].

The function of Kns1 proteins in filamentous fungi remains unknown. To investigate the cellular function of LAMMER kinase Kns1 in *M. oryzae*, we cloned *KNS1*, a homolog of *S. cerevisiae KNS1*. Deletion of *KNS1* resulted in defects in pathogenicity of the rice blast fungus. Additionally, Kns1 was necessary for the expression of genes involved in the DNA damage response. Furthermore, deletion of *MoKNS1* affected TOR activity and autophagy. We propose that MoKns1 in *M. oryzae* is important for its pathogenicity, partly through regulating autophagy and the DNA damage pathway for invasion.

## 2. Results

### 2.1. MoKns1 Is a Homolog of Kns1 in Saccharomyces cerevisiae

We used the Kns1 peptide sequence from budding yeast as the query in a BlastP search within the *M. oryzae* genome database in EnsembIFungi (http://fungi.ensembl.org/Magnaporthe_oryzae/Info/Index) (accessed on 20 July 2018). We identified a Kns1 homolog (MGG_16904) that shared 51% amino acid similarity with Kns1, which we named MoKns1. MoKns1 showed high homology with its orthologues in other fungi (Figure 1), including *FgKNS1* of *Fusarium graminearum* (53% identity), *CoKNS1* of *Colletotrium orbiculare* (67% identity), *HsKNS1* of *Homo sapiens* (53% identity), *MmKNS1* of *Mus musculus* (46% identity), *NcKNS1* of *Neurospora crassa* (58% identity). MoKns1 encodes a LAMMER kinase comprising 704 amino acids, which contains a conserved LAMMER motif. Our results revealed that Kns1 is well conserved across various organisms including yeast, filamentous fungi, plants, mammals and Homo sapiens (Figure 1).

### 2.2. MoKns1 Is Required for Conidiation and Virulence

To investigate the role of MoKns1 in *M. oryzae*, a single mutant of *KNS1* was constructed by ATMT. The *MoKNS1* gene deletion was confirmed by PCR and relative expression levels analysis (Appendix A). The complemented transformant strain Δ*Mokns1–C* was generated by transforming the Δ*Mokns1* mutant with a genetic construct containing *MoKNS1* fused to green fluorescent protein (*MoKNS1*–GFP), as shown in Appendix A. To investigate the biological roles of MoKns1 in *M. oryzae*, vegetative growth, conidiation and pathogenicity were assayed by comparison to the wild-type 70−15 strain, the ∆*Mokns1* mutant and the complementation strain ∆*Mokns1–C*. As shown in Figure 2A,B, the ∆*Mokns1* strain grew more slowly than the wild-type 70−15 and ∆*Mokns1–C* strains on complete medium (CM) plates during 9 days of incubation. The mycelial morphology of the wild-type 70−15 strain and the ∆*Mokns1* mutant was not much different (Appendix A). In addition, the ∆*Mokns1* mutant produced almost no conidia (Figure 2C), as indicated also by the conidiophore morphology (Appendix A). To examine the effect of *MoKNS1* on virulence, we inoculated mycelial plugs onto barley leaves and found that lesions caused by the Δ*Mokns1* mutant hardly spread (Figure 2A). The rice spray experiment could not be completed because almost no conidia were produced by the ∆*Mokns1* mutant. These data demonstrated that *MoKNS1* plays vital roles in conidiation and pathogenicity in *M. oryzae*.

Unlike other higher eukaryotes, which have several isoforms of LAMMER kinase, *M. oryzae* has only one LAMMER kinase ortholog. Kns1 contains a so-called LAMMER motif, EHLAMMEAVVD, within the subdomain X of the kinase catalytic domain [10]. Considering the domains of MoKns1, we artificially divided MoKns1 into three segments, the first segment included a Disordered region (1–131 aa), the second segment included a second Disordered region (151–272 aa), and the third segment included the LAMMER protein kinase domain (332–670 aa) (Figure 2D). The three segments were complemented into the Δ*Mokns1* mutant to obtain Δ*Mokns1–C^N1^*, Δ*Mokns1–C^N2^*, Δ*Mokns1–C^N3^*, respectively. We found that the growth and conidiation of the three segment-supplemented strains could not be restored by supplementing any single segment, and their growth and conidiation were almost the same as those of the Δ*Mokns1* mutant. The pathogenicity of the three segment-supplemented strains was still very high and basically was recovered to its initial levels (Figure 2A–C). Then, two segments were complemented into the Δ*Mokns1* mutant to obtain Δ*Mokns1–C^N1+N2^*, Δ*Mokns1–C^N2+N3^*, Δ*Mokns1–C^N1+N3^*. The growth and conidiation of any of the two-segment-supplemented strains could not be restored, and the growth and conidiation were almost the same as those of the Δ*Mokns1* mutant. The pathogenicity of any of the two-segment-supplemented strains was still very high and basically was recovered to its initial levels (Figure 2A–C). These results indicated that the growth and conidiation of MoKns1 required all amino acid sequences, but any isolated sequence affected the pathogenicity of *M. oryzae*.

### 2.3. MoKns1 Is Required for the Response to DNA Replicative Stress

To elucidate the cellular functions of MoKns1 in *M. oryzae*, the response of the Δ*Mokns1* mutants to several stresses was investigated. All Δ*Mokns1* mutants showed sensitivity to MMS- and HU-caused DNA replicative stress during the S-phase (Figure 3A–D). Under DNA replicative stress, both fission and budding yeast activate the expression of specific genes for DNA repair and cell cycle control [17]. We homologously aligned the genes in yeast and examined the expression levels of *MoMSH2* (encoding a DNA mis-match repair factor), DNA damage checkpoint genes (*MoRAD17*, *MoCDS1*, *MoCHK1*), Mitotic exit (*MoCDC14*, *MoCDC15*, *MoDBF2*, *MoNUD1*, *MoMOB1*, *MoCDC5*, *MoTEM1*), anaphase-promoting complex genes (*MoAPC3*, *MoAPC1*, *MoCDC20*) by qRT−PCR. The Δ*Mokns1* strain showed obviously increased expression levels of *MoMSH2*, mitotic exit genes (*MoCDC14*, *MoCDC15*, *MoDBF2*, *MoNUD1*, *MoMOB1*, *MoCDC5*, *MoTEM1*) and anaphase-promoting complex genes (*MoAPC3*, *MoAPC1*, *MoCDC20*). However, the expression levels of the DNA damage checkpoint genes (*MoRAD17*, *MoCDS1*, *MoCHK1*) in the Δ*Mokns1* mutant were significantly lower than those in the wild-type 70−15 strain (Figure 3E). These results suggest that MoKns1 plays a role in the transcriptional regulation of DNA replicative relative genes and is required for the response to DNA replicative stress.

### 2.4. MoKns1 Interacts with MoAtg18 and Is Involved in Autophagy 

To understand the mechanism of action of MoKns1, a yeast two-hybrid screening assay was conducted with an *M. oryzae* cDNA library. MoAtg18 was identified as a candidate that interacted with MoKns1. First, the interaction between full-length MoKns1 and MoAtg18 was not detected by the two-hybrid assay. Since MoKns1–N3 contains a LAMMER motif, we examined the interaction between MoKns1–N3 and MoAtg18. As shown in Figure 4A, Kns1–N3 interacted with MoAtg18. Subsequently, we examined whether MoKns1–N1 and MoKns1–N2 interacted with Atg18. We found that MoKns1–N2 interacted with MoAtg18 (Figure 5A), but the interaction between MoKns1–N1 and Atg18 was not detected. To further confirm the authenticity of the interaction, we used pull-down experiments to verify the interaction between MoKns1–N2, MoKns1–N3 and MoAtg18 (Figure 4B and Figure 5B). Our results showed that MoKns1–N2 and MoKns1–N3 interacted with MoAtg18.

Autophagy is emerging as an important process for virulence of pathogenic fungi [22]. TOR is a critical modulator of the autophagy process [5]. To test whether MoKns1 regulates the activity of MoTor, the relationship between MoKns1 and MoTor kinase was verified after treatment with rapamycin, a specific inhibitor of the Tor kinase. Compared with the wild type 70−15 strain, the Δ*Mokns1* mutant was more sensitive to rapamycin (Figure 6A,B). To examine whether the autophagy process in the Δ*Mokns1* mutant was impaired, we tagged MoAtg8 at its N-terminal with GFP on the chromosomes in the wild-type and Δ*Mokns1 M. oryzae* mutant to monitor localization and degradation of GFP−MoAtg8 by western blot. These strains were starved in the SD−N medium for different durations. The delivery of GFP−MoAtg8 to the vacuoles in Δ*Mokns1* mutant was significantly increased (2 h: 47%, 4 h: 52% vacuoles with GFP fluorescence) (Figure 6C,D). These results suggest that the degradation of GFP−MoAtg8 was accelerated in the Δ*Mokns1* mutant. Then, the autophagy flux was detected in the wild-type 70−15 strain and the Δ*Mokns1* mutant. In the presence of nutrients, autophagy was more intense, and a more intense MoAtg8−PE (autophagy marker to monitor autophagy flux) band was found in the Δ*Mokns1* mutant (Figure 6E). We concluded that MoKns1 is involved in autophagy in *M. oryzae*. *M. oryzae* contains many ATG genes. These ATG genes, i.e., Atg1–9, 11–18, 20, 22, 22–24, 26–29, affect the pathogenicity of *M. oryzae* [18,21,23,24,25,26,27,28]. Transcription analysis of these genes was performed by qRT−PCR. Only the three genes *MoATG4*, *MoATG14*, *MoATG29* showed lower expression, and the expression of *MoATG29* gene was significantly downregulated in the Δ*Mokns1* mutant. The expression levels of other ATG genes were all upregulated, and the *MoATG2*, *MoATG5*, *MoATG6*, *MoATG11*, *MoATG15*, *MoATG17*, *MoATG22*, *MoATG2* genes were significantly upregulated (Figure 7).

## 3. Discussion

*Magnaporthe oryzae* causes rice blast disease via a penetration peg that is produced by the appressorium and its expansive growth across cells [29]. These invasion processes are regulated by a variety of signal pathways. Protein kinases play important roles in many living activities. Studies have shown that many protein kinases affect the pathogenicity of *M. oryzae* [30]. KNS1 is distantly related to members of the CDC28/cdc2 gene family [8]. LAMMER kinase was found to be very conserved and has autophosphorylation activity [31]. In yeast, TOR regulates ribosome and tRNA synthesis via the LAMMER kinase [9]. However, there is little research on the functions of Kns1 proteins in filamentous fungi. In order to explore the function of Kns1 in rice blast, we deleted the *MoKNS1* gene in the wild-type strain 70−15 and systematically analyzed the effects of this deletion on the growth, development and pathogenicity of *M. oryzae*. In this work, we characterized MoKns1 in *M. oryzae* and showed that it is required for fungal development and plant pathogenicity. We discovered that *MoKNS1* is involved in autophagy and interacts with Atg18, which was discovered in the rice blast fungus. Additionally, we also found that *MoKNS1* participates in maintaining DNA replication and regulates TOR activity. To the best of our knowledge, these results have not been reported for other species and will help us better understand the various functions of Kns1 protein in plant pathogenic fungi.

Autophagy is an essential, conserved self-eating process that cells perform to allow the degradation of intracellular components. The complete process of autophagy consists of three steps: induction, autophagosome formation and autophagosome−lysosome fusion and degradation [32]. The pathogenicity of *M. oryzae* was seriously affected by the deletion of autophagy-related genes. In yeast, Atg18 localizes to both the preautophagosomal structure (PAS) and to punctate structures at the vacuolar membrane [33]. Vacuolar localization of Atg18 is also dependent on the transmembrane protein Vac7 [34]. PAS-localized Atg18 is involved in the localization of the autophagy proteins Atg2 and Atg9. Atg18 is essential for vesicle formation in both autophagy and the cytoplasm-to-vacuole targeting (Cvt) pathway. The *atg18* mutant affects the autophagy process [33]. In *M. oryzae*, the pathogenicity of the Δ*Moatg18* mutant was completely lost, and autophagy was blocked. The evolutionarily conserved protein kinase Tor (mechanistic target of rapamycin) is a crucial cell growth and autophagy regulator that integrates nutritional cues [19]. It has been reported that excessive stimulation of autophagy through constant inhibition of MoTor activity hinders host plant infection by *M. oryzae* [35]. Therefore, the precise regulation of MoTor activity may be responsible for the modulation of autophagy. In our study, the Δ*Mokns1* mutant was more sensitive to rapamycin. We inferred that MoKns1 might act as a novel autophagy inhibition factor to repress autophagy by regulating MoTor activity.

DNA replication is the biological process that produces two identical DNA replicas from one original DNA molecule. DNA replication occurs in all living organisms and is the most important process in biological inheritance. In *M. oryzae*, proteins involved in DNA replication are also important for development. Early appressorium development requires the nucleus in the germinating conidial cell to undergo DNA replication. In Δ*bimE* mutants, arrested cell cycle progression results in loss of pathogenicity, suggesting that mitosis is required for fungal infection [36]. In *M. oryzae*, the MoCds1 kinase is an S-phase checkpoint of the DNA damage response (DDR) pathway. Control of DNA replication and modulation of turgor are necessary for infecting plant cells [37]. In our study, we found a significant difference in the expression of DNA synthesis- and mitosis-related genes and higher sensitivity to the DNA synthetic inhibitors HU and MMS in the Δ*Mokns1* mutant.

In summary, we demonstrated that MoKns1 is involved in autophagy and characterized its biological functions. MoKns1 plays a crucial role in hyphal growth, conidiation, invasive growth and pathogenicity in *M. oryzae*. In addition, the Δ*Mokns1* mutant appeared to be sensitive to the DNA synthetic inhibitors HU and MMS. In the Δ*Mokns1* mutant, we found a significant difference in the expression of DNA synthesis- and mitosis-related genes. Furthermore, we provide new insights on how MoKns1 negatively regulates autophagy by interacting with MoAtg18.

## 4. Methods and Materials

### 4.1. Strains and Culture Conditions

*M. oryzae* strain 70–15 was used as the wild-type strain. All strains were cultured on solid CM for growth at 25 °C for 6–9 days. The mycelia were harvested from the liquid CM and used for genomic DNA and RNA extractions. Please refer to previous references for specific culture conditions [38].

### 4.2. Gene Deletion and Complementation and Plasmid Constructions

To construct *the MoKNS1* knockout vector, the 5′ untranslated region (UTR) and the 3′ UTR of *MoKNS1* were cloned with the primers *KNS1*–5F/5R and *KNS1*–3F/3R (Appendix A), then ligated into the EcoRI/HindIII sites of pKO1B with a bar resistance gene. The resultant knockout vector was introduced into the wild-type strain 70−15 by Agrobacterium-mediated transformation (ATMT) [39]. All the primers used in the gene deletion experiment are listed in Appendix A. The obtained mutants were screened by PCR. EcoRΙ and BamHΙ were used to double slice the PKD5 vector. The complement DNA fragment that contained the entire *MoKNS1* gene driven by the native promoter was amplified by PCR and ligated to GFP by using the primers KNS1-GFP-F/R. To construct plasmids expressing MoKns1–N1–AD (1–131aa), MoKns1–N2–AD (151–272aa), MoKns1–N3–AD (332–670aa), MoKns1–BD, we used the related cDNA fragments from the *M. oryzae* cDNA. To construct the plasmids used in the pull-down experiments, we cleaved the pGEX4T−1 vector using the restriction enzyme EcoRΙ, then the coding sequence of MoKns1–N2 was amplified with KNS1–N2–GST–F/R primers. The GST–MoKNS1–N2 vector was obtained by homologous recombination linking the amplified MoKns1–N2 fragment with the cut pGEX4T–1 vector. The His−MoAtg18 vector was obtained by homologous recombination linking the amplified His−MoAtg18 fragment with the cut pET21a vector (HindIII).

MoAtg18 was thus fused with GST and His. All constructs were generated via homologous recombination (pEASY−Basic Seamless Cloning and Assembly Kit; Transgen Biotech, Beijing, China).

### 4.3. Pathogenicity Assay

Mycelial plugs of the 70−15, Δ*Mokns1* and Δ*Mokns1–C* strains were inoculated on detached barley, placed in an incubator at a temperature of 25 °C with a light/dark cycle of 16/8 h for 4 days.

### 4.4. Quantification of Gene Expression by qRT−PCR

Total RNA samples were extracted from the mycelia using the PureLinkTM RNA mini kit (Vazyme) according to the manufacturer’s protocol. The reverse transcriptase HiScript III RT SuperMix for qPCR (Vazyme) was used to prepare cDNA. qRT−PCR was run on a Lightcycler96 (Roche) Real−time PCR System with chamQ Universal SYBR qPCR Master Mix (Vazyme). The 2^−^^ΔΔCT^ method was used to calculate the relative quantity of each transcript using the *M. oryzae* actin gene as the internal control. The experiment was repeated three times with three biological replicates. The primers used are listed in Appendix A.

### 4.5. Autophagy Assays

The strains targeted with GFP−MoAtg8 were inoculated in liquid CM medium at 25 °C for 36 h, then shifted to SD−N medium for 2 h and 4 h to induce autophagy. Mycelium treated with the MM−N medium was assessed for the degradation of GFP−MoAtg8 by immunoblotting assays with anti-GFP (rabbit; 1:10,000; Abcam, UK) and anti-GAPDH antibodies (rabbit; 1:5000; Huaan). For the MoAtg8 and the MoAtg8−PE turnover assays, the MoAtg8 and the MoAtg8−PE bands were detected using an anti-Atg8 antibody (1:2000, BML) with 13.5% SDS−PAGE.

### 4.6. Sensitivity Test to Chemical Agents and Physical Stresses

For the stress assays, mycelial blocks were inoculated on the CM medium with 0.02% MMS, 20 mM HU, 100 ng/mL Rapamycin at 25 °C and photographed at day 8. Please refer to the previous references for the specific experimental steps [35].

### 4.7. Yeast Library Screen and Yeast Two-Hybrid Assay

A yeast library screen was conducted according to the yeast screening library manual of Clontech company. The target decoy gene fragment MoKNS1 was inserted into thepGBKT7 vector, transferred into E. coli and expanded to extract the MoKns1–AD plasmid. Competent yeast cells were prepared according to the experimental operation manual (Clontech, San Francisco, CA, USA). After adding MoKns1–AD and cDNA−BD library plasmids of M. oryzae into the yeast competent cells, we spread them on SD–Ade–Leu–Trp medium for 3–5 days. Then, yeast single spots were selected for PCR verification, and the PCR products were sequentially examined by YouKang Biological Company.

Y2H assays were performed using the Matchmaker Yeast Two-Hybrid System (Clontech) following the manufacturer’s instructions. The cDNA of MoKns1–N1, MoKns1–N2, MoKns1–N3, MoAtg18 was cloned into the AD and BD vectors. The recombinant plasmids were co-introduced into AH109 cells, and cell growth examined on SD–Leu–Trp–His–Ade and SD–Leu–Trp media. The pGADT7–T/pGBKT7–53 (AD–T/BD–53) plasmid was used as a positive control, and pGADT7/pGBKT7(AD/BD) was used as a negative control. All primers are listed in Appendix A.

### 4.8. In Vitro Pull-Down Assays

The cDNA of *MoKNS1* was cloned and inserted into the vector pGEX4T−1 to obtain the GST–MoKns1–N2 and GST–MoKns1–N3 plasmids. The full−length cDNA of MoAtg18 was fused with the vector pET21a to acquire the His−MoAtg18 plasmid. The resulting plasmid DNA was introduced into *Escherichia coli* strain BL21 (DE3+Plyse) cells. Then, proteins were extracted from these cells using a lysis buffer (150 mM NaCl, 10 mM Tris−HCl (pH 7.5), 0.5% Triton X−100 and 0.5 mM EDTA). Whole cell lysates were analyzed by SDS−PAGE and transferring to a Coomassie blue solution for staining to confirm the expression of the resultant proteins. To ensure normal protein expression, 100 μL of glutathione agarose beads (Smart-Lifescience, Changzhou, China) was incubated with GST, GST–MoKns1–N1 or GST–MoKns1–N2 proteins at 4 °C for 2 h. Then, the cells were incubated with the bacterial lysate containing His−MoAtg18 at 4 °C for 2 h. The eluted proteins were detected by immunoblotting with anti-GST and anti-His antibodies (HUABIO, Hangzhou, China).

### 4.9. Determination of Conidiation

The wild type 70−15, Δ*Mokns1* mutant and Δ*Mokns1–C* strains were cultured on solid CM medium at 25 °C for 8 days. The edge of colonies were selected with a hole punch, and the mycelial plugs were transferred to a CM petri dish of 9 cm; each strain was transferred to 5 petri dishes and cultured at 25 °C for 11 days. Then, 5 mL sterile water was added to each plate, the mycelium was gently scraped off with a sterilized coating stick, and then the conidia were filtered into a 50 mL centrifuge tube using three layers of filter paper. After centrifugation at 7500 rpm for 10 min, the supernatant was discarded, and the conidia were collected. Finally, the conidia were suspended again in 30 mL sterile water, and the number of conidia was counted by a blood counting plate under a microscope. Each sample was examined 6 times, and the results were recorded.

## Figures and Tables

**Figure 1 ijms-23-08104-f001:**
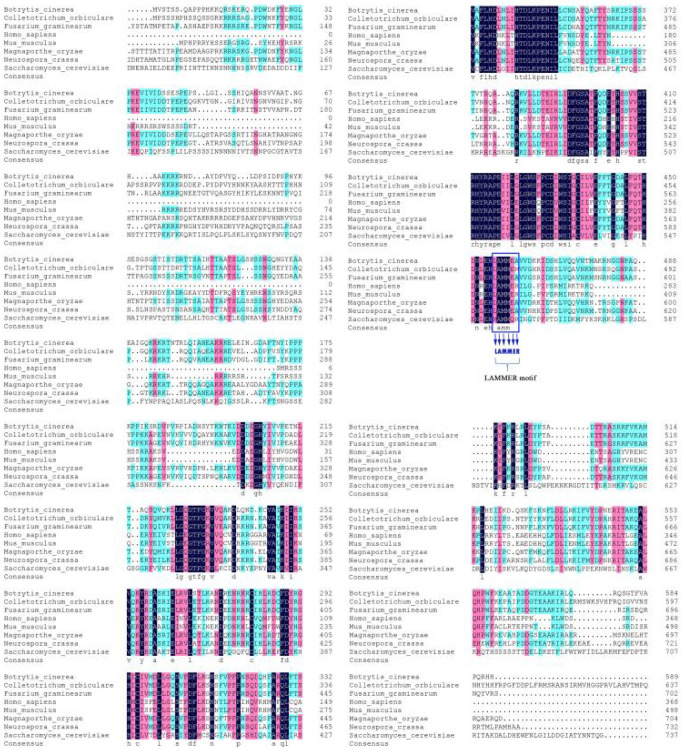
Sequence alignment of the amino acids of Kns1 in different species. A. The sequence alignment of Kns1 amino acids was made by the DNAMAN 8. The compared sequences were from *Magnaporthe oryzae* (MoKns1), *Colletotrichum orbiculare* (Lkh1, TDZ24089), *Fusarium graminearum* (Kns1, EYB28111.1), *Homo sapiens* (6FYK_A), *Mus musculus* (NP_001156904.1), *Neurospora crassa* (XP_957701.3), *Saccharomyces cerevisiae* (CAD6636070.1). All these Kns1 proteins have a conserved LAMMER motif (blue framed part).

**Figure 2 ijms-23-08104-f002:**
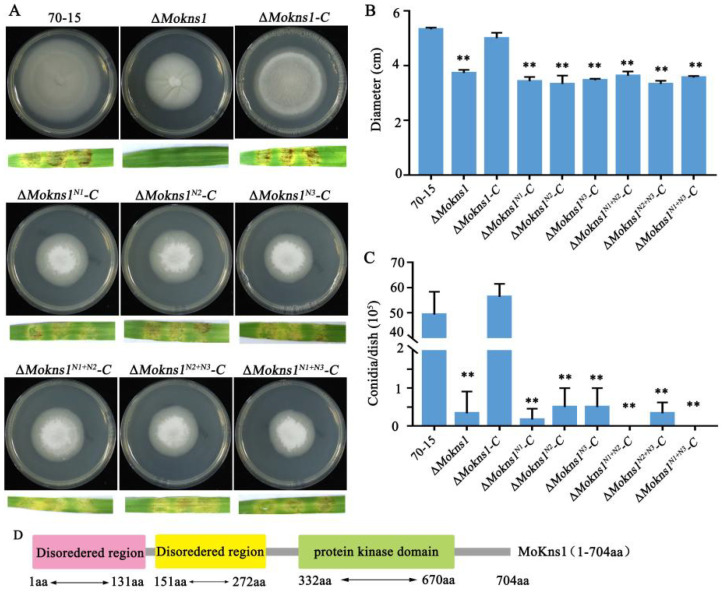
MoKns1 is important for growth, conidiation and pathogenicity. (**A**). The wild-type 70−15 strain, the Δ*Mokns1* mutant and the Δ*Mokns1^N1^–C*, Δ*Mokns1^N2^–C*, Δ*Mokns1^N3^–C*, Δ*Mokns1^N1+N2^–C*, Δ*Mokns1^N2+N3^–C*, Δ*Mokns1^N1+N3^–C*, Δ*Mokns1–C* complemented strains were grown on CM medium for 9 days. (**B**). Statistical analysis of the diameters of hyphae from the WT, the Δ*Mokns1* mutant and the Δ*Mokns1^N1^–C*, Δ*Mokns1^N2^–C*, Δ*Mokns1^N3^–C*, Δ*Mokns1^N1+N2^–C*, Δ*Mokns1^N2+N3^–C*, Δ*Mokns1^N1+N3^–C*, Δ*Mokns1–C* complemented strains on CM (Ducan’ test, ** *p* < 0.01). (**C**). Statistical analysis of conidia production (Ducan’ test, ** *p* < 0.01). (**D**). Map of MoKns1 with three domains.

**Figure 3 ijms-23-08104-f003:**
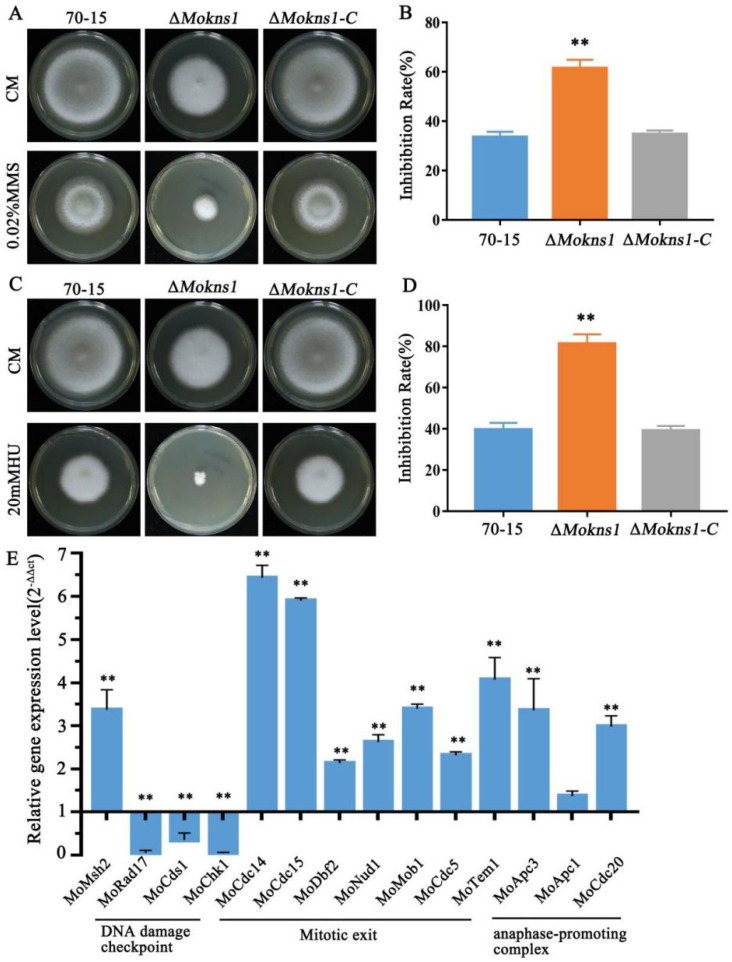
MoKns1 is involved in the DNA damage stress response. (**A**,**C**), sensitivity test for DNA damage stress using hydroxyurea (HU) and methyl methanesulfonate (MMS). The strains were incubated on CM supplemented with 20 mM HU or 0.02% MMS at 28 °C for 8 days. (**B**,**D**), the inhibition rate was determined by plotting the percentage of colonies in the presence of various stresses in comparison with the control. Asterisks denote statistical significances (*p* < 0.01). (**E**). Relative mRNA levels of DNA damage checkpoint genes (*MoMSH1*/MGG_00879, *MoRAD17*/MGG_17052, *MoCDS1*/MGG_04790, *MoCHK1*/MGG_03729), mitotic exit genes (*MoCDC14*/MGG_04637, *MoCDC15*/MGG_04100, *MoDBF2*/MGG_02757, *MoNUD1*/MGG_02395, *MoMOB1*/MGG_03151, *MoCDC5*/MGG_09960, *MoTEM1*/MGG_04896), anaphase-promoting genes (*MoAPC3*/MGG_17195, *MoAPC1*/MGG_03314, *MoCDC201*/MGG_01236) were quantified using qPCR, and the results for the aerial mycelia of the ∆*Mokns1* mutant were normalized to α-ACTIN expression and compared to the results for the WT. Error bars represent standard deviation. Significant differences compared with the wild-type strain were estimated using Duncan’s test (** *p* < 0.01).

**Figure 4 ijms-23-08104-f004:**
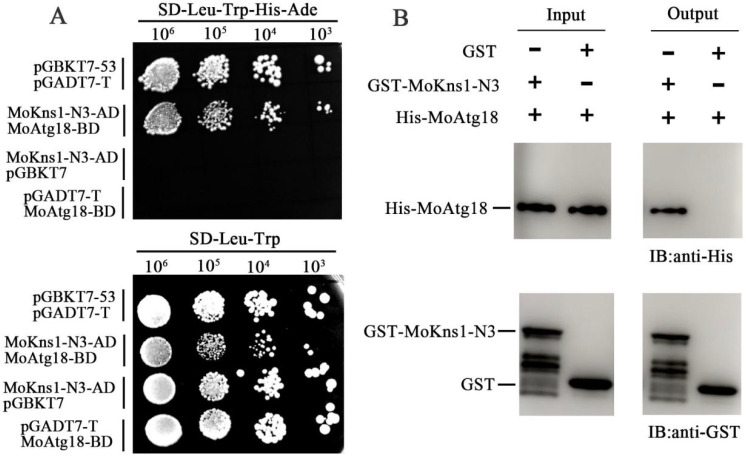
MoKns1–N3 interacts with MoAtg18. (**A**). Yeast two−hybrid analysis. The pair of plasmids pGBKT7−53 and pGADT7−T was used as the positive control. Plates were incubated at 30 °C for 3 days before being photographed. (**B**). Pull down assays. The recombinant GST–MoKns1–N3 or GST bound to glutathione Sepharose beads was incubated with *E. coli* cell lysate containing His–MoAtg18. Eluted proteins were analyzed by immunoblot (IB) with the monoclonal anti–His and anti−GST antibodies.

**Figure 5 ijms-23-08104-f005:**
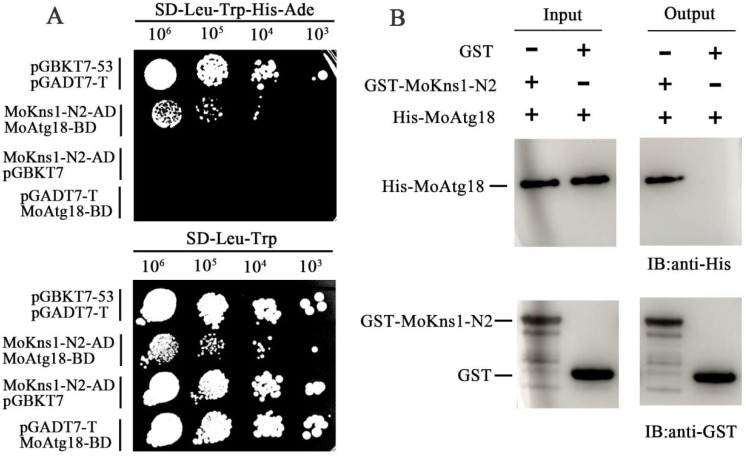
MoKns1–N2 interacts with MoAtg18. (**A**). Yeast two-hybrid analysis. The pair of plasmids pGBKT7−53 and pGADT7−T was used as the positive control. The plates were incubated at 30 °C for 3 days before being photographed. (**B**). Pull-down assays. Recombinant GST–MoKns1–N2 or GST bound to glutathione Sepharose beads was incubated with an *E. coli* cell lysate containing His−MoAtg18. The eluted proteins were analyzed by immunoblotting (IB) with the monoclonal anti−His and anti−GST antibodies.

**Figure 6 ijms-23-08104-f006:**
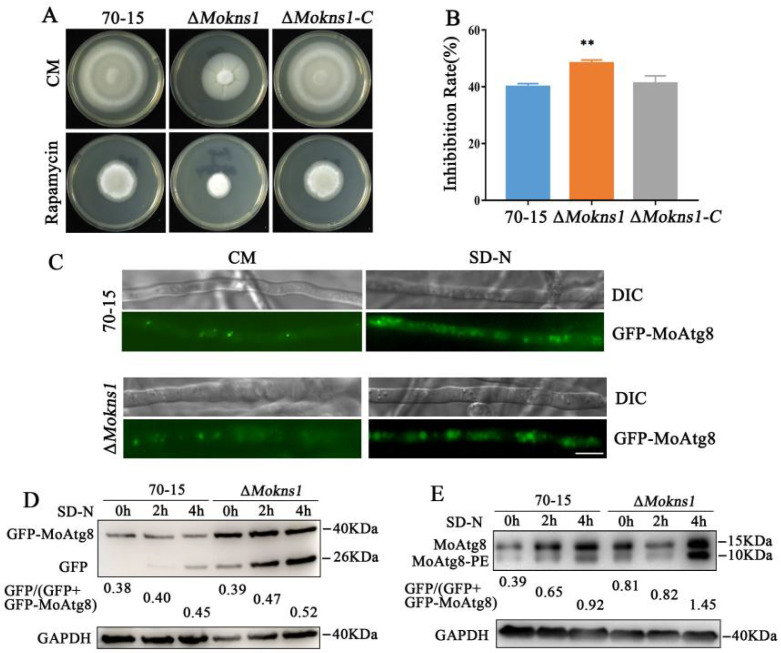
MoKns1 is involved in autophagy. (**A**). Sensitivity test to Rapamycin (Duncan’s test, ** *p* < 0.01). (**B**). The inhibition rate was determined by comparing the percentage of colonies in the presence of Rapamycin to the percentage of colonies in the control. Asterisks denote statistical significances (*p* < 0.01). (**C**). The localization of GFP−MoAtg8 in the 70−15 strain and the Δ*Mokns1* mutant. The strains were cultured in liquid CM medium for 2 days, then transferred to SD−N medium for 4 h. Bar, 10 μm. (**D**). Immunoblot analysis of the degradation of GFP−MoAtg8 in SD−N medium for 2 h and 4 h in the WT strain and Δ*Mokns1* mutant. (**E**). Immunoblot analysis of MoAtg8/MoAtg8−PE turnover in the wild-type 70−15 strain and Δ*Mokns1* mutant.

**Figure 7 ijms-23-08104-f007:**
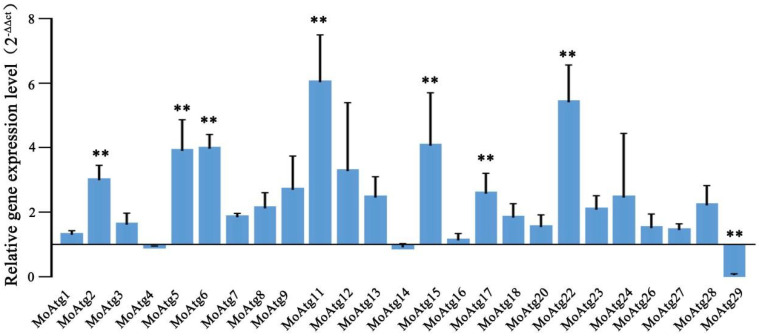
Analysis of the expression levels of core autophagy genes in *M. oryzae*. The relative mRNA levels of core autophagy genes (*MoATG1*/MGG_06393, *MoATG2*/MGG_16734, *MoATG3*/MGG_17909, *MoATG4*/MGG_03580, *MoATG5*/MGG_09262, *MoATG6*/MGG_03694, *MoATG7*/MGG_07297, *MoATG8*/MGG_01062, *MoATG9*/MGG_09559, *MoATG11*/MGG_04486, *MoATG12*/MGG_00598, *MoATG13*/MGG_454, *MoATG14*/MGG_03698, *MoATG15*/MGG_12828, *MoATG16*/MGG_05255, *MoATG17*/MGG_07667, *MoATG18*/MGG_03139, *MoATG20*/MGG_12832, *MoATG22*/MGG_09904, *MoATG23*/MGG_10579, *MoATG24*/MGG_03638, *MoATG26*/MGG_03459, *MoATG27*/MGG_02386, *MoATG28*/MGG_08061, *MoATG29*/MGG_02790) were quantified using qPCR, and the results were normalized to α-ACTIN expression and compared to the WT results in the aerial mycelia of ∆*Mokns1* mutant. Error bars represent standard deviation. Significant differences compared with the WT were estimated using Duncan’s test (** *p* < 0.01).

## Data Availability

Data are contained within the article or Appendix A.

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
