# Peer review of "The LAMMER Kinase MoKns1 Regulates Growth, Conidiation and Pathogenicity in Magnaporthe oryzae"

_ijms, 2022, doi:10.3390/ijms23158104_

Round 1
Reviewer 1 Report
In my opinion, the manuscript submitted for review is of a good scientific standard and its individual parts are consistent. I like the very idea of the manuscript, the research and the descriptive part, however, the graphic part needs to be refined - the fig. 1 is of poor quality. I do not know if it is the fault of 1) the conversion from ms word to PDF? 2) or the poor quality of the original Fig.. I ask the authors to improve these parts of the manuscript.
Author Response
We really appreciate your professional comments. We changed Figure 1 with high quality.

Reviewer 2 Report
In this study, the authors attempted to identify a unique LAMMER kinase MoKns1 in M. oryzae and then investigate its cellular function and found its roles in the growth, conidiation, and pathogenicity of M. oryzae. The authors then provided several observations to demonstrate that MoKns1 regulates autophagy and DNA damage response pathways, thereby affecting pathogenicity. In general, the scientific aim of this study sounds, and the investigation may provide novel insights and further support the emerging vision of the vital roles of autophagy in fungal pathogenesis. However, more supportive evidence with closer looks must be included, further detailed methods should be provided, and substantial modifications in written text and its organization should be carefully revised to make the investigation thorough and meet a professional standard for publication.
Here are major comments:
1. In the “Figure S1. Identification of knockout mutants”: Which primers were used for screening? Please provide a strategy map for creating deletion and complementation strains. PCR screening is absolutely insufficient to verify a genetically modified strain, please provide southern blot data or transcriptional expression and/or protein expression data for the succeeded targeted genetically modified strains. Also, what was ATMT method (line 88)? Please additionally provide a brief description of ATMT method used in this study.
2. Please additionally provide the specific method used to conduct the virulence test in this study.
3. Please additionally provide information on the method and procedure used to conduct immunoblotting assays in this study.
4. Line 154-155 the authors stated that “the ∆Mokns1 strain grew more slowly than the wild type 70-15 and MoKNS1-C strains on complete medium (CM) plates after 9 days of incubation”, how’s about their hyphal morphologies?
5. Please additionally provide the method for conidiation assay used in this study. Further, in line 156 the authors stated that “∆Mokns1 mutant produce almost no conidia (Figure 2C)”, this needs more clarification since data was represented by 10^5, meaning that there were about 10^2-10^3 conidia produced, thus it may not almost no conidia as the statement. Also, this reviewer wondered if certain other specific conidium-induced media were also conducted before drawing this conclusion? More importantly, a closer observation of conidiophore morphologies and deficiency must be conducted to obtain in-depth supportive data. Also, the transcriptional expression of representative genes related to conidiation will be compelling evidence to support this statement.
6. In the “Figure 1”: There was only one part in this figure, so A was not necessary. Further, why a full alignment was not presented? Based on the number of amino acids indicated, it looks like that certain parts from the initial and ending parts were deleted?
7. In the “Figure 2”: Information in the text (“9 days” in line 155) and figure legend (“8 days” in line 164) was different. Also, there was no figure 2D but this was written in the text (line 158).
8. Line 167-187: how could the authors define “disordered region”? As results of these important regions covered a reasonable space in this investigation, an outline map of MoKns1 with those domains should be additionally included to help ease for readers. Also, how’s about the supplement with double segments? These would provide more insights and confirm the robust function of each specific segment.
9. How was the transcriptional expression level of MoKNS1 gene over the course of M. oryzae life cycle (i.e., hypha, conidia, other stages)? Given that the authors successfully created MoKNS1-GFP strain (line 150), this reviewer wondered how was the expression of MoKns1 protein over the course of M. oryzae life cycle through showing its fluorescent signal of subcellular localization on each specific stage?
10. Line 212-213: The authors stated that “a yeast two-hybrid screening assay was conducted with an M. oryzae cDNA library”, however, there was no screening method or obtained data presented in this manuscript. Please provide more informative clarification of what experiments have examined and the detailed consequences of the examined yeast two-hybrid screening assay in general and in particular. Also, how MoAtg18 was identified as a candidate for the interacting partner of MoKns1? Since it was interesting and seemed to be in conflict with the authors’ statement in lines 215-216 that: “the interaction between full-length MoKns1 and MoAtg18 was not detected by two-hybrid assay”. If the pull-down assay was not determined for MoKns1-N1 and MoAtg18? Further, what was the authors’ perspective on the obtained interaction results? If the MoKns1-N1 was the inhibitory factor for the no-interaction of full-length with MoAtg18? This scenario should be thoroughly addressed in the discussion section.
11. Figure 6C-D: These results must be included the close microscopic observations of GFP localized into the vacuoles and its quantitive intensity data to support those immunoblotting results.
12. The reference was error presented and needed to be revised.
Author Response
Thank you very much for your professional advice. We have made changes according to your comments.
